# Clinical Outcome after Surgical Treatment of Sacral Chordomas: A Single-Center Retrospective Cohort of 27 Patients

**DOI:** 10.3390/cancers16050973

**Published:** 2024-02-28

**Authors:** Stavros Goumenos, Georgios Kakouratos, Ioannis Trikoupis, Panagiotis Gavriil, Pavlos Gerasimidis, Konstantinos Soultanis, Pavlos Patapis, Vasileios Kontogeorgakos, Panayiotis Papagelopoulos

**Affiliations:** 11st Department of Orthopaedic Surgery, “Attikon” University General Hospital, National and Kapodistrian University of Athens, 1 Rimini Street, 12461 Athens, Greece; stgoumenos@gmail.com (S.G.); kakourade@hotmail.com (G.K.); giannistrikoupis@gmail.com (I.T.); gavriilpan@gmail.com (P.G.); pavlosgerasimidis@gmail.com (P.G.); ksoultanis@otenet.gr (K.S.); vaskonto@gmail.com (V.K.); 23rd Department of Surgery, “Attikon” University General Hospital, National and Kapodistrian University of Athens, 1 Rimini Street, 12461 Athens, Greece; ppatapis@med.uoa.gr

**Keywords:** chordoma, sacrectomy, recurrence, wound-related complications, risk factors

## Abstract

**Simple Summary:**

Sacral chordomas are extremely rare tumors. The aim of our retrospective study was to assess the survivorship of patients with sacral chordoma who underwent en-bloc surgical excision and to investigate potential risk factors for tumor recurrence and postoperative surgical site complications. The estimated 5-year disease-free survivorship was 53.9%. The tumor size, use of plastic reconstructive techniques, duration of the surgery, ligation of iliac vessels, and serum albumin levels were associated with postoperative wound-related compilations, while surgical margins were associated with local recurrence. Despite the high complications rate, patient survivorship, after the surgical excision of sacral chordomas, was not impaired.

**Abstract:**

**Introduction:** The aims of our study were (1) to determine disease-specific and disease-free survival after the en-bloc resection of sacral chordomas and (2) to investigate potential risk factors for tumor recurrence and major postoperative wound-related complications. **Methods:** We retrospectively analyzed 27 consecutive patients with sacral chordomas who were surgically treated in our institution between 2004 and 2022. Three patients (11.1%) had a recurrent tumor and four patients (14.8%) had history of a second primary solid tumor prior to or after their sacral chordoma. A combined anterior and posterior approach, colostomy, plastic reconstruction, and spinopelvic instrumentation were necessitated in 51.9%, 29.6%, 37%, and 7.4% of cases, respectively. The mean duration of follow-up was 58 ± 41 months (range= 12–170). Death-related-to-disease, disease recurrence, and major surgical site complications were analyzed using Kaplan–Meier survival analysis, and investigation of the respective risk factors was performed with Cox hazard regression. **Results:** The estimated 5-year and 10-year disease-specific survival was 75.3% (95% CI = 49.1–87.5%) and 52.7% (95% CI = 31–73.8%), respectively. The estimated 1-year, 5-year, and 10-year disease-free survival regarding local and distant disease recurrence was 80.4% (95% CI = 60.9–91.1%), 53.9% (95% CI = 24.6–66.3%), and 38.5% (95% CI = 16.3–56.2%), respectively. The mean survival of the recurred patients was 61.7 ± 33.4 months after their tumor resection surgery. **Conclusions:** Despite the high relapse rates and perioperative morbidity, long-term patient survival is not severely impaired. Positive or less than 2 mm negative resection margins have a significant association with disease progression.

## 1. Introduction

Chordomas are a rare malignancy of notochordal origin accounting for 1–4% of malignant bone tumors and for over 50% of all primary tumors of the sacrum [1,2].

Wide en-bloc resection is the surgical gold standard for the management of these slow-growing, locally invasive, chemo- and radio-resistant tumors, with the extent of the resection being the most important predictor for local recurrence and a favorable prognosis [1,3].

The surgical treatment of sacral chordomas remains a challenging task often requiring combined surgical approaches, complex soft tissue reconstruction, adjuvant radiotherapy, and a multidisciplinary team approach [3]. Even so, death-related-to-disease and local recurrence rates are often exceeding 27% and 43%, respectively, while the postoperative surgical site complication rates after sacrectomy remain notoriously high [3,4]. The early identification and modification of the implicated risk factors for disease progression and treatment-related complications can prove to be critical in selecting the most appropriate therapy strategy and optimizing patient clinical and functional outcomes.

The aims of our study were to determine disease-specific and disease-free survival after the en-bloc resection of primary or recurrent sacral chordomas (primary endpoint) and also to investigate potential risk factors for tumor recurrence and postoperative wound-related complications in our cohort (secondary endpoint).

## 2. Methods

### 2.1. Study Design

After obtaining institutional board approval, we retrospectively analyzed data from 27 consecutive patients with primary or recurrent sacral chordomas that were surgically treated in our institution between January 2004 and April 2022.

Exclusion criteria were (1) patients under the age of 16 years, (2) sacral tumors of non-notochordal origin, (3) chordomas of the mobile spine, upper cervical spine, or any other anatomical region, and (4) patients with primary or recurrent chordomas of the sacrum who denied or were not eligible for surgical treatment (e.g., metastatic or extensive local disease). All patients in our cohort were diagnosed and treated by the same multidisciplinary team of our specialized orthopedic oncology department.

Initial preoperative planning included contrast-enhanced lumbosacral MRI followed by guided core needle biopsy and CT scan disease staging. All surgeries were carried out by our interdisciplinary surgical team, which consists of orthopedic oncology and spine surgeons, as well as abdominal and plastic surgeons. After being discharged, patients were repetitively evaluated by our orthopedic oncology outpatient clinic at the following time points: 3, 6, 9, 12, 18, 24, 36, 48, and 60 months after their tumor resection surgery. The mean follow-up in our cohort was 58 ± 41 months (range= 12–170 months).

### 2.2. Patients

Out of the 27 patients surgically treated for sacral chordoma, 17 were men (63%) and 10 were women (37%). The mean ± SD of age, BMI (Body Mass Index), and age-adjusted CCI (Charlson Comorbidity Index) were 62.6 ± 11.9, 28.9 ± 4.6, and 5.3 ± 1.66, respectively. There were 3 patients (11.1%) referred to our institution with recurrent sacral chordoma initially surgically treated at another hospital and 24 patients (88.9%) with primary chordoma of the sacrum. Four patients (14.8%) had a history of a second primary solid tumor besides their sacral chordoma, which involved the thyroid, stomach, uterus, and breast. The primary symptoms reported at the time of presentation were local pain exacerbated when sitting (85.2%), followed by neurological deficits, including bladder or bowel dysfunction or radiculopathies (55.6%), and finally a palpable mass (29.6%). None of the patients who planned to undergo tumor resection surgery had known metastatic disease at the time of diagnosis. Serum albumin levels were 4 ± 0.7 g/dL and 2.5 ± 0.4 g/dL at the final preoperative measurement and the first postoperative day, respectively. The size of the tumor was calculated preoperatively using the known formula for measuring a sphere’s volume (*V* = 4πr^33, *r* = mean value of the three radius measurements of the sphere-like-shaped tumor in the coronal, sagittal, and axial plane in the preoperative MRI scan) and hence had a mean volume of 621.8 ± 647.8 cm^3^. Extensive dorsal soft tissue involvement was documented in 16/27 patients (59.3%) based on the MRI report. Four patients (14.8%) had undergone preoperative radiotherapy for local disease control before their tumor resection surgery [5].

### 2.3. Surgical Procedure

A combined anterior and posterior approach was performed in 14/27 patients (51.9%), whereas only a midline posterior approach was performed for the remaining 48.1% based on the most cephalad extent of the tumor (Figure 1). 

In total, 8 patients (29.6%) required a colostomy, and 10/27 patients (37%) required flap coverage due to extensive posterior soft tissue deficit, 5/10 of which were bilateral gluteal advancements, and another 5/10 were pedicled transpelvic vertical rectus abdominis myocutaneous flaps supplied by the inferior epigastric vessels (VRAM) (Figure 2).

Ipsilateral or bilateral internal iliac vessel ligation was performed in 14/27 cases (51.9%). All VRAM flaps and 3/5 gluteal flaps were mobilized in patients with internal iliac ligation. The sacral resection level was usually placed at half or one sacral vertebra above the tumor. Total sacrectomy (Type 1A) with spinopelvic fixation was performed in only 1 patient (3.7%), subtotal sacrectomy above the level of the S1 foramen (1B) was performed in 6/27 patients (22.2%), 1 of which also underwent spinopelvic fixation, and, finally, subtotal sacrectomy below the level of the S1 foramen (1C) was performed in the remaining 20/27 patients (74.1%). Among the 1C group, 9/27 (33.3%) sacrectomies extended to the level of S1–S2 intervertebral disk, 8/27 (29.6%) sacrectomies extended through the S2 level or at the level of S2–S3 intervertebral disk, and 3/27 (11.1%) sacrectomies were below the S3 level. There were no hemisacrectomies in this cohort [6] (Figure 3).

The average duration of the surgeries was 410 ± 208 min. The tumor resection margins were negative with more than >2 mm of healthy tissue around the tumor and intact tumor capsule (N > 2 mm) in 16/27 patients (59.3%), negative with less than 2 mm of healthy tissue around the tumor and intact tumor capsule (N < 2 mm) in 8/27 patients (29.6%) and positive (P) in 3/27 patients (11.1%) [7,8]. The mean maximum area of the resected specimen was 228 ± 107.9 cm^2^ based on the histopathology report.

### 2.4. Risk Factors

The investigated potential risk factors for local or distant disease recurrence (LDR or DDR) and surgical site complications included patient demographics (age, gender, BMI, CCI), preoperative and postoperative serum albumin levels, prior resection surgery at the site (primary or recurrent chordoma), size of the tumor measured in their preoperative MRI, dorsal soft tissue involvement, level of sacrectomy, duration of surgery, surgical approach implemented, ligation of internal iliac vessels, need for colostomy, surgical excision margins, maximum specimen area, need for plastic reconstruction, and use of preoperative radiotherapy.

### 2.5. Statistical Analysis

Continuous variables were expressed as mean ± standard deviation (SD), whereas categorical variables and rates within groups were expressed as percentages.

Disease-specific survival (DSS), disease-free survival (DFS), and wound-related complications (WRCs) were analyzed as time-to-event outcomes utilizing survival models. Specifically, the cumulative probability of death-related-to-disease, disease progression, and WRCs was estimated through Kaplan–Meier curves, and comparisons between groups were made using the Log-rank test. The starting point was the date of surgery, and the end point was the date of the first event of interest (death, relapse, complication) or the last follow-up. All potential risk factors for disease recurrence or WRCs were initially analyzed through univariate survival analysis, and those factors that were found to have a statistically significant correlation with these events (*p* < 0.05) were pooled together to a Cox hazard regression model to investigate their independent association with those outcomes. All tests were two-sided, and all confidence intervals were set to 95%. Statistical analysis was performed using SPSS v.23^®^.

## 3. Results

### 3.1. Disease Mortality

The overall survival rate in our cohort was 63% (17/27 patients) during a mean follow-up of 58 ± 41 months (range = 12–170 months). Five patients (18.5%) died within the first 5 years after their surgery from reasons related to their disease, followed by three additional patients (11.1%) that died after 5 years postoperatively. All eight related-to-disease deaths (29.6%) occurred within 7 years after their tumor resection surgery. The estimated cumulative 5-year and 10-year DSS after Kaplan–Meier analysis was 75.3% (95% CI = 49.1–87.5%) and 52.7% (95% CI = 31–73.8%), respectively (Figure 4).

The mean survival of the recurred patients was 61.7 ± 33.4 months after their tumor resection surgery.

### 3.2. Disease Recurrence

Regarding disease recurrence, 12/27 patients (44.4%) relapsed during the entire follow-up of 58 ± 41 months (range= 12–170 months). More specifically, 4/27 patients (14.8%) had an LDR, 2/27 (7.4%) were later diagnosed with distant metastases, and 6/27 (22.2%) had both. All recurrences occurred within 6 years after surgery and only 2/12 (16.7%) after the first 5 years postoperatively. The estimated cumulative 1-year, 5-year, and 10-year DFS regarding LDR and DDR after Kaplan–Meier analysis was 80.4% (95% CI = 60.9–91.1%), 53.9% (95% CI = 24.6–66.3%), and 38.5% (95% CI = 16.3–56.2%), respectively (Figure 5).

All 10 patients with LDR were subjected to radiotherapy, and 5 of them were planned for revision tumor excision surgery [9]. Adjuvant chemotherapy (imatinib) was administered to 3/10 relapsed cases [10].

In terms of the analyzed risk factors for disease recurrence, the size of the tumor, a N < 2 mm or P surgical resection, dorsal soft tissue infiltration by the tumor, and a recurrent chordoma showed statistically significant correlations with disease recurrence (*p* = 0.038, *p* < 0.001, *p* = 0.022, and *p* = 0.001, respectively). After multivariate Cox regression, only the surgical excision margins maintained a significant independent association with the risk for disease recurrence (HR = 10.18, 95% CI: 2.03–50.94, *p* = 0.005) (Table 1).

### 3.3. Wound-Related Complications (WRCs)

Overall, there were 24 postoperative complications in 20/27 patients (74.1%). The most common postoperative complications were wound-related (18/22) and were further divided into minor superficial wound healing disorders (6/18) and major surgical site complications (12/18), which included four partial flap necrosis in patients with iliac vessel ligation (two VRAM and two gluteal flaps) and eight wound dehiscence events that required deep surgical debridement under general anesthesia. In total, 7/12 cases (58.3%) were associated with infection. The remaining 6/22 postoperative complications included 2 sacral stress fractures, 1 spinopelvic instrumentation failure, 1 pulmonary embolism, 1 myocardial infarction, and 1 bowel obstruction.

All surgical site complications occurred within 2 months after tumor resection surgery. The estimated risk for overall WRCs and major WRCs after tumor resection surgery ± subsequent plastic reconstruction was 66.7% (95% CI = 47.8–81.4%) and 53.6% (95% CI = 36.5–75.5%), respectively, after Kaplan–Meier analysis.

After univariate survival analysis, we found that an increased size of the tumor, an increased maximum area of the resected specimen, the need of plastic reconstructive surgery, an increased duration of surgery, the ligation of internal iliac vessels, and decreased serum albumin levels on the first postoperative day displayed a statistically significant impact on the estimated risk for major WRCs (*p* = 0.016, *p* = 0.013, *p* = 0.039, *p* = 0.012, *p* = 0.003 and *p* = 0.001, respectively). However, when subjected to multivariate Cox regression, none of those risk factors proved to have an independent association with the risk for a major WRC event (Table 2).

We planned for the 4/10 patients who suffered failure of their flaps to undergo revision plastic reconstruction surgery. The remaining 8/12 patients with major WRCs underwent at least one deep soft tissue debridement under general anesthesia ± subsequent vacuum-assisted secondary wound closure treatment (VAC), whereas those with merely minor wound dehiscence underwent only one superficial debridement with subsequent primary wound closure under local anesthesia. The patient with spinopelvic instrumentation failure underwent a revision fixation (Figure 3), and the two patients with sacral stress fractures were percutaneously fixated with screws.

### 3.4. Final Evaluation

Concerning the final functional outcome of the patients based on their last clinical evaluation, 22/27 patients (81.5%) required intermittent urinary bladder self-catheterization, 20/27 (74.1%) reported sexual dysfunction, and 11/27 (40.1%) required bowel training. In terms of locomotion status, 3/27 patients (11.1%) required a wheelchair, 7/27 (25.9%) were mobile with the use of one cane, and the remaining 17/27 (63%) were fully mobile without the use of any assistance.

## 4. Discussion

The successful surgical management of sacral chordomas remains quite challenging and requires an experienced multidisciplinary team approach as these tumors have a high propensity for LDR, postoperative WRCs, and permanent genitourinary impairment [3,4]. Hence, optimizing the oncological and functional outcome after sacral chordoma resection is of paramount importance, as is also identifying and modifying the implicated risk factors for DFS and WRCs before the treatment strategy for en-bloc excision and adjuvant radiotherapy is commenced.

An important limitation of our study is that of being a retrospective cohort, which entails many difficulties regarding data retrieval from the patients’ medical records, even though the majority of the data were collected in a prospective manner. Also, it lacks a control group, being for instance a survivorship comparison between surgically and conservatively treated sacral chordomas with advanced radiotherapy modalities [9,11]; however, proton and carbon beam radiotherapy are unfortunately not available for use in our country. Moreover, our sample of patients is small, but this is usually the case when it comes to these rare and highly specific case series, and our minimum follow-up was reduced to 1 year after tumor resection surgery. Therefore, larger prospective randomized case–control trials are needed to further delineate the impact of all those risk factors on disease progression and WRCs when it comes to these challenging and highly heterogenic cases, but they are also needed to shed light on the ideal management of recurrent disease.

As for the primary endpoint of our study, the 5-year DSS and 5-year DFS in our cohort were 75% and 54%, respectively. These high relapse rates are indicative of the natural history of these tumors and are consistent with numerous cohort studies of various specialized orthopedic oncology centers, which demonstrate 5-year DSS and 5-year DFS rates of 70–92% and 48–71%, respectively [7,8,12,13,14,15,16,17,18,19,20,21,22] (Table 3). Lee IJ et al. demonstrate a 5-year and 10-year DSS of 70% and 47% in a USA-population-based analysis of 473 chordomas of the pelvis [20], whereas Houdek MT et al. report 10-year DSS and 10-year DFS rates of 72% and 58%, respectively, in a large multicenter cohort of 193 patients with sacral chordomas [19]. According to the USA National Cancer Database, the 5-year overall survival of 942 patients after treatment for sacrococcygeal chordomas was 65.3% [23].

In our cohort, the tumor resection margins had a significant and independent impact on DFS after Cox hazard regression. Many authors present similar results, with the surgical excision margins being the most widely accepted risk factor for DFS and DSS [4,5,7,8,12,13,15,18,19,21,22,24,25], followed by recurrent tumor en-bloc resection surgeries [4,12,13,24], the size of the tumor [7,15,17,19,22,23,24], and the surrounding soft tissue infiltration [21,22,26]. Kayani B et al. report median survival of 23, 67, and 90 months after sacral chordoma resection in patients with gluteus maximus invasion, piriformis invasion, and no dorsal soft tissue invasion, respectively, though without increasing the risk for LDR [22].

Patients with clear surgical margins primarily display a significantly lower risk for LDR [19] and a higher 5-year survival than those with an inadequate resection margin (86% versus 67%) in a large cohort enrolling 115 patients with sacral chordomas who underwent surgical treatment [12]. However, a gross R0 total excision of a spinal/sacral chordoma cannot be easily obtained due to the ventral proximity of the tumor to vital structures, but even so, suboptimal resection still provides a significant benefit in patient survival compared to nonoperative treatment [23,27]. A cut-off of 1.5 mm of surrounding healthy tissue and an intact tumor capsule have been associated with significantly higher DSS and DFS regarding marginal resections [7,8].

Furthermore, a tumor size greater than 8–9 cm in diameter has been correlated with increased risk for DDR and death-related-to-disease [19,20,22], whereas a tumor diameter greater than 10 cm entails a 5-year survival of 54.8% [23]. On the other hand, the most cephalad extent of the tumor within the sacrum, which has also been reported to be a significant negative predictor for DFS and DSS in numerous studies [7,12,19,21], was not found to play an important impact in our cohort.

In particular, delving into the risk factors for LDR versus DDR Kerekes D et al. concludes that wide excision margins and adjuvant therapy are associated with a lower rate of both LDR and DDR, whereas prior tumor surgeries also significantly correlate with increased LDR rates [4]. Similarly, van Wulfften Palthe ODR et al. found that not receiving adjuvant radiation was a significant and independent risk factor for LDR in the primary setting, while an increased tumor size had a significant impact on LDR, DDR, and overall patient survival [17]. However, our limited number of patients was not suitable for such an extensive subgroup analysis.

Inadequate resection margins and the tumor level within the sacrum were also found to be significant and independent risk factors for re-recurrence after the surgical excision of recurred sacral chordomas in this highly specific cohort of 31 patients by Yang Y et al. [28]. The authors report 5-year and 10-year survival rates of 67.3% and 53.9%, respectively, and a re-recurrence rate of 37.5% compared to the 100% in our study (3/3 patients).

Finally, 4/27 of our patients had another primary solid tumor in their medical history. This finding is quite interesting and adds up to the prevailing concept of increased overlap rates between various types of cancers and musculoskeletal tumors and definitely requires further investigation [29].

Regarding the early postoperative complications after such a challenging tumor resection surgery, WRCs are by far the most common events [4,16,17,18,19,22,30,31,32,33,34], with reported rates up to 57.8% [35]. Two large cohorts of patients with sacral chordomas from Mayo Clinic report 68% overall complication rates and 32% WRC rates after sacrectomy [19,36]. In a recent systematic review of 13 analyzed studies and 384 sacrectomies, Branco E Silva M et al. report reoperations rates of 28% due to WRCs [30].

In a large cohort of 87 patients who underwent VRAM reconstruction surgery subsequently to sacrectomy, Houdek MT et al. report overall complications, wound dehiscence, and deep surgical site infection rates of 79%, 47%, and 26%, respectively, and they conclude that preoperative radiotherapy and obesity were significant predictors for both wound dehiscence and deep surgical site infection, but the size of the tumor was not [32]. However, in another cohort of 42 sacrectomies for chordoma, Schwab JH et al. support that rectus abdominus flaps are associated with decreased WRCs [13]. Detailed description of the surgical techniques and respective complications of the various flaps utilized after sacrectomy for chordoma has been previously provided by Deskoulidi P et al. [37].

Apart from the need for plastic reconstructive surgery, we also identified the duration of surgery, low postoperative serum albumin levels, the ligation of internal iliac vessels, the increased size of the tumor, and the increased area of the resected specimen as significant risk factors for major WRCs after the univariate survival analysis. These parameters have not been extensively studied, and the literature regarding risk factors for WRCs after sacral chordoma resection is quite limited. Chen KW et al. demonstrate a significant and independent impact of preoperative serum albumin levels < 3.0 g/dL, operation time > 6 h, and previous surgeries on the risk for surgical site infection after multivariate logistic regression in 45 sacral chordoma resections [31]. On the other hand, Ruggieri P et al. do not find a correlation between the size of the tumor, the level of the resection, nor any previous resection surgeries with increased risk for deep surgical site infection in a cohort of 82 chordomas and giant cell tumors of the sacrum [33]. However, although the size of the sacral tumor was not found to be a significant negative predictor for complication-free patient recovery in neither of the previous studies, larger tumors (dimension ≥ 9 cm and volume ≥ 500 cm^3^) have generally been associated with postoperative wound complications, as has a high sacral level resection [19]. In a large retrospective cohort of 387 patients who underwent sacrectomy for sacral tumors, Li D et al. report WRCs of 29.2%, almost half of which (13.2%) were associated with infection. The authors conclude that maximum tumor diameter > 10 cm, history of diabetes, use of instrumentation, previous radiotherapy, rectum rupture, and CSF leakage are significant and independently associated risk factors for WRCs after multivariate logistic regression. The last three of those factors, in addition to longer duration of surgery, were also significant and independent predictors of deep surgical site infection following sacrectomy [34]. Unfortunately, the aforementioned univariately significant risk factors in our study were not independently associated with major WRCs after multivariate Cox regression, most probably due to our small number of patients, but also because of them being powerful confounding factors with the need for plastic reconstructive surgery.

Although the few existing relevant studies have not implicated the ligation of the internal iliac vessels as a significant contributor to WRCs [32,38], the present study showed that those patients had a significantly higher risk for major surgical site complications (10/14, 71.4%). Li D et al. also found a significant univariate correlation between internal iliac artery embolic occlusion and postoperative noninfectious wound complications in their large cohort, but this association was not maintained through their multivariate analysis [34]. Finally, neo-and/or adjuvant radiotherapy, which has been associated with postresection surgical site sequalae and sacral stress fractures [17,19,32,34], did not elicit a significant risk for major WRCs in our cohort.

## 5. Conclusions

Despite the occurrence of high relapse rates and perioperative morbidity after sacral chordoma resection, long-term patient survival was not severely impaired. A recurrent tumor, the surgical excision margins, tumor size, and dorsal soft tissue infiltration by the tumor have a significant impact on disease recurrence. Furthermore, the increased size of the tumor, increased maximum area of the resected specimen, the need for plastic reconstructive surgery, increased duration of surgery, ligation of internal iliac vessels, and decreased serum albumin levels on the first postoperative day also seem to play a significant role in the already high risk for major surgical site complications after sacrectomy for chordoma.

## Figures and Tables

**Figure 1 cancers-16-00973-f001:**
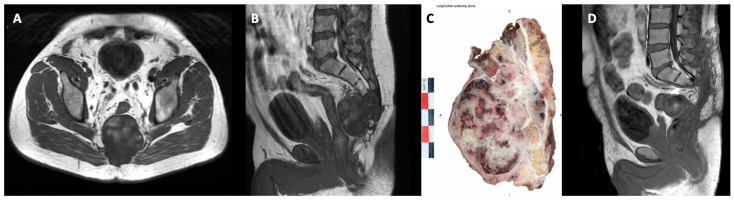
(**A**,**B**): T1-weighted axial and sagittal MRI of an S3 level sacral chordoma (male, 55 year), (**C**) post-en-bloc resection specimen (sagittal midline view), (**D**) postoperative progression-free MRI evaluation at 6 months after tumor resection surgery (contrast-enhanced T1-weighted sagittal view).

**Figure 2 cancers-16-00973-f002:**
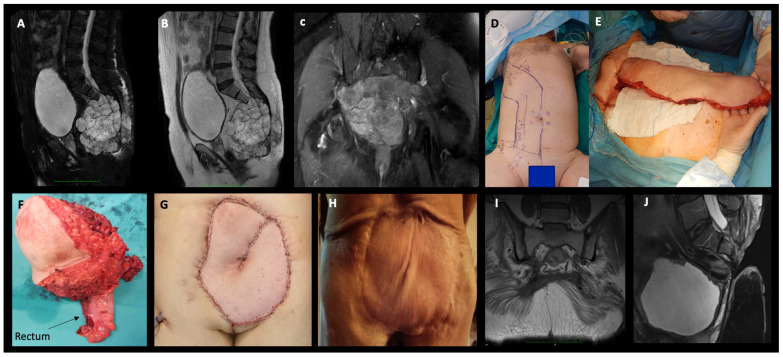
(**A**,**B**) T2-weighted with fat suppression and T2-weighted MRI sagittal views showing an S2 level sacral chordoma (male, 62 year), (**C**) postcontrast T1-weighted with fat suppression coronal MRI view showing dorsal soft tissue and gluteus maximus infiltration by the tumor (**D**,**E**) intraoperative photographs showing preparation and harvest of the VRAM flap, (**F**) photograph of the resected specimen showing en-bloc resection of the tumor with the skin and part of the rectum attached (combined approach), (**G**) restoration of the soft tissue defect with the VRAM flap, (**H**) satisfactory wound healing result without complications 1 year postoperatively, (**I**,**J**) T1-weighted coronal MRI view and T2-weighted with fat suppression sagittal MRI view, respectively, showing no local recurrence at 1 year postoperative follow-up.

**Figure 3 cancers-16-00973-f003:**
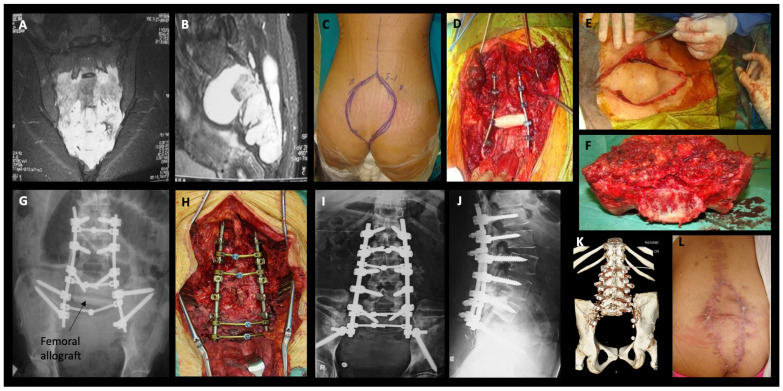
(**A**,**B**) T2-weighted with fat suppression coronal and sagittal MRI views of an S1 level sacral chordoma (female, 58 year), (**C**,**D**) en-bloc total sacrectomy and spinopelvic fixation (combined approach), (**E**) restoration of the soft tissue defect with a VRAM flap, (**F**) en-bloc postresection specimen of the entire sacrum, (**G**) early postoperative AP plain radiograph of the lumbar spine and pelvis showing the lumbo-iliac fixation and reconstruction with a femoral allograft, (**H**) the patient underwent spinopelvic revision surgery due to instrumentation failure, (**I**,**J**) anteroposterior and lateral plain radiographs of lumbar spine, (**K**) postoperative 3D volume rendering CT image of the revised lumbo-iliac fixation, (**L**) satisfactory wound healing result 1 year postoperatively. The patient remains disease free at 14 years postoperatively.

**Figure 4 cancers-16-00973-f004:**
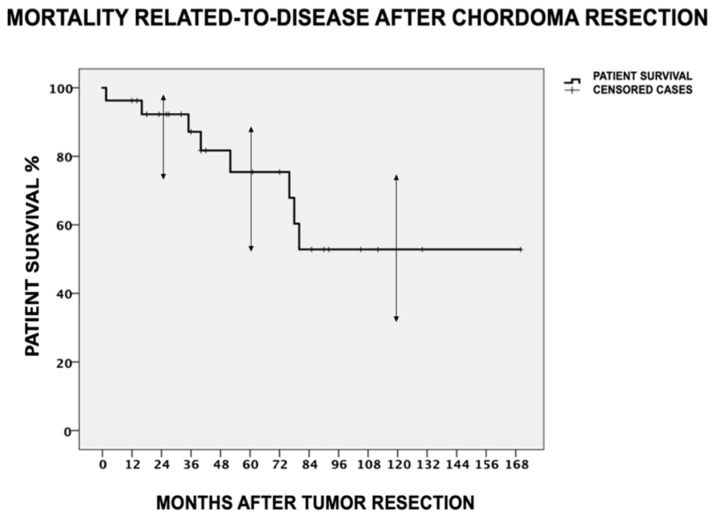
Related-to-disease mortality after sacral chordoma resection. The estimated cumulative 5-year and 10-year disease-specific survival after Kaplan–Meier analysis was 75.3% (95% CI = 49.1–87.5%) and 52.7% (95% CI = 31–73.8%), respectively. All related-to-disease deaths occurred within 7 years after tumor resection surgery. The mean survival of the recurred patients was 61.7 ± 33.4 months after their tumor resection surgery.

**Figure 5 cancers-16-00973-f005:**
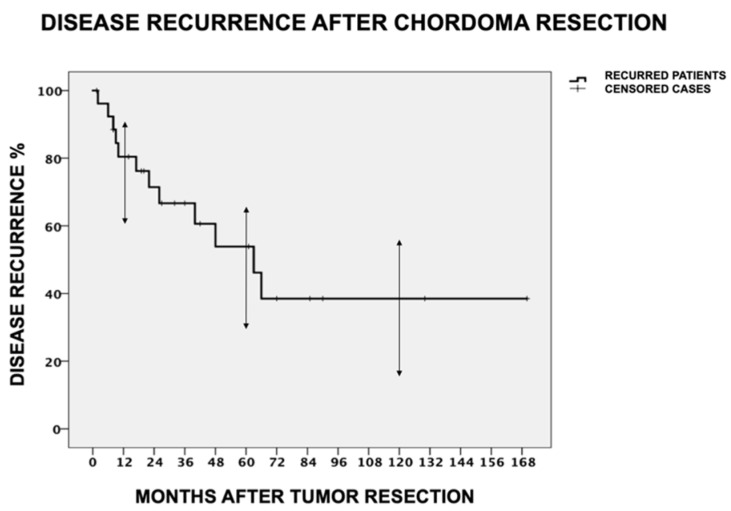
Disease recurrence incidence after sacral chordoma resection. The estimated cumulative 1-year, 5-year, and 10-year disease-free survival regarding local and distant disease recurrence after Kaplan–Meier analysis was 80.4% (95% CI = 60.9–91.1%), 53.9% (95% CI = 24.6–66.3%), and 38.5% (95% CI = 16.3–56.2%), respectively. All recurrences occurred within 6 years after surgery. The mean time-to-recurrence for the recurred patients was 26.4 ± 22.5 months after tumor resection surgery.

**Table 1 cancers-16-00973-t001:** Risk factors for disease recurrence after sacral chordoma resection.

	Survivors (N = 15 Patients)	Recurred (N = 12 Patients)	Significance (Univariate)	Significance (COX)	Hazard Ratio (95% CI)
Survival (months)	55 ± 47.1 [2:170]	61.7 ± 33.4 [14:112]	*p* = 0.411	-	-
Age (years)	60.7 ± 14.7 [36:87]	64.9 ± 6.8 [57:78]	*p* = 0.667	-	-
BMI (kg/m^2^)	28.2 ± 4.4 [22:35]	29.8 ± 5.1 [21:36]	*p* = 0.236	-	-
Gender (M/F)	10 (67%)/5 (33%)	7 (58%)/5 (42%)	*p* = 0.541	-	-
CCI	5.2 ± 2 [2:9]	5.4 ± 1.1 [4:7]	*p* = 0.732	-	-
Primary/recurrent tumor	15 (100%)/0 (0%)	9 (75%)/3 (25%)	*p* = 0.001 *	*p* = 0.457	0.255 (0.007–9.335)
Second primary tumor (yes/no)	2 (13%)/13 (87%)	2 (17%)/10 (83%)	*p* = 0.358	-	-
Tumor size (cm^3^)	500.5 ± 456.9 [5:1440]	787.2 ± 839.5 [34:3050]	*p* = 0.038 *	*p* = 0.321	1.001 (0.999–1.003)
Dorsal soft tissue involvement (yes/no)	7 (47%)/8 (53%)	9 (75%)/3 (25%)	*p* = 0.022 *	*p* = 0.518	0.438 (0.036–5.355)
Level of sacrectomy	9 (60%) S1–S2/ 4 (27%) S2–S3/ 2 (13%) S3–S4	8 (67%) S1–S2/ 3 (25%) S2–S3/ 1 (8%) S3–S4	*p* = 0.854	-	-
Surgical approach (combined/posterior)	9 (60%)/6 (40%)	5 (42%)/7 (58%)	*p* = 0.645	-	-
Surgical margins (N > 2 mm/N < 2 mm/P)	14 (93%)/ 1 (7%)/ 0 (0%)	2 (17%)/ 7 (58%)/ 3 (25%)	*p* < 0.001 *	*p* = 0.005 *^a^*	10.179 (2.034–50.941)
Preoperative radiotherapy (yes/no)	1 (7%)/14 (93%)	3 (25%)/9 (75%)	*p* = 0.173	-	-

Continuous variables are given as mean ± standard deviation with the range in brackets, whereas categorical variables are given as a number with the respective percentage in parentheses. A recurrent chordoma, the surgical excision margins, the size of the tumor, and dorsal soft tissue infiltration by the tumor showed significant correlations with disease recurrence (*). After multivariate Cox regression, the resection margins maintained a significant association with the risk for chordoma recurrence (*^a^*). Abbreviations: BMI = Body Mass Index, CCI = Charlson Comorbidity Index. Surgical resection margins: N > 2 mm (negative and more than 2 mm), N < 2 mm (negative but less than 2 mm), and P (positive).

**Table 2 cancers-16-00973-t002:** Risk factors for major wound-related complications after sacral chordoma resection.

	Uncomplicated (N = 15 Patients)	Complicated (N = 12 Patients)	Significance (Univariate)	Significance (COX)	Hazard Ratio (95% CI)
Age (years)	64.1 ± 13.5 [36:87]	60.6 ± 9.6 [38:75]	*p* = 0.702	-	-
BMI (kg/m^2^)	28.3 ± 4.2 [22:34]	29.5 ± 5.3 [20:35]	*p* = 0.273	-	-
Gender (M/F)	9 (60%)/6 (40%)	8 (67%)/4 (33%)	*p* = 0.829	-	-
CCI	5.2 ± 1.8 [2:9]	5.4 ± 1.5 [3:8]	*p* = 0.584	-	-
Primary/recurrent tumor	15 (100%)/0 (0%)	9 (75%)/3 (25%)	*p* = 0.114	-	-
Tumor size (cm^3^)	329 ± 310.7 [5:950]	1102.7 ± 796 [385:3050]	*p* = 0.016 *	*p* = 0.608	0.001 (0.998–1.001)
Dorsal soft tissue involvement (yes/no)	7 (47%)/8 (53%)	9 (75%)/3 (25%)	*p* = 0.081	-	-
Maximum specimen area (cm^2^)	172 ± 96.6 [70:380]	297.9 ± 77.8 [220:425]	*p* = 0.013 *	*p* = 0.874	0.002 (0.977–1.027)
Level of sacrectomy	7 (47%) S1–S2/5 (33%) S2–S3/ 3 (20%) S3–S4	10 (83%) S1–S2/2 (17%) S2–S3/ 0 (0%) S3–S4	*p* = 0.053	-	-
Duration of surgery (min)	364 ± 164.3 [150:665]	535 ± 231 [370:830]	*p* = 0.012 *	*p* = 0.841	−0.026 (0.755–1.257)
Iliac vessel ligation (yes/no)	4 (27%)/11 (73%)	10 (83%)/2 (17%)	*p* = 0.003 *	*p* = 0.574	1.058 (0.72–115.14)
Colostomy (yes/no)	3 (20%)/12 (80%)	5 (42%)/7 (58%)	*p* = 0.209	-	-
Flap (VRAM/Gluteal/No)	1 (7%)/2 (13%)/ 12 (80%)	4 (33%)/3 (25%)/ 5 (42%)	*p* = 0.039 *	*p* = 0.924	0.149 (0.054–24.85)
Preoperative serum albumin (G/DL)	3.9 ± 0.7 [2.8:4.7]	4.2 ± 0.8 [3:4.9]	*p* = 0.215	-	-
Postoperative serum albumin (G/DL)	2.8 ± 0.2 [2.4:3.2]	2.2 ± 0.3 [1.7:2.7]	*p* = 0.001 *	*p* = 0.122	−2.472 (0.004–1.929)
Preoperative radiotherapy (yes/no)	1 (7%)/14 (93%)	3 (25%)/9 (75%)	*p* = 0.155	-	-

Continuous variables are given as mean ± standard deviation with the range in brackets, whereas categorical variables are given as a number with the respective percentage in parentheses. The size of the tumor, maximum specimen area, duration of the surgery, ligation of iliac vessels, use of a flap, and postoperative serum albumin levels showed significant correlations with major wound-related postoperative complications (*). However, those were not maintained after multivariate Cox regression.

**Table 3 cancers-16-00973-t003:** Survival and Disease Progression Rates after Sacral Chordoma Resection.

Study	N	5 y DSS	5 y DFS	Risk Factors for Disease Progression	Treatment	WRCs
Fujiwara et al., 2019 [7]	48	88%	46%	RM, aRT, tumor size	Surgical ± aRT	-
Ji T et al., 2017 [12]	115	81%	52%	RM, recurrent tumor	Surgical ± aRT	-
Schwab et al., 2009 [13]	42	77%	56%	Prior and intralesional resections	Surgical ± aRT	45%
Radaelli et al., 2016 [15]	99	92%	62%	RM, tumor size	Surgical ± aRT	-
Van Wulfften et al., 2018 [17]	101	65–79%	64–86%	aRT, tumor size	Surgical ± aRT	40%
Fuchs et al., 2005 [18]	52	74%	59%	RM, age	Surgical ± aRT	33%
Houdek et al., 2019 [19]	193	75%	71%	RM, RT dose, tumor size, excision level	Surgical ± aRT	32%
Kayani et al., 2015 [22]	58	62%	52%	RM, tumor size, differentiation, aRT, infiltration of sacroiliac joint/muscles	Surgical ± aRT	17%
Varga et al., 2015 [24]	167	≈70%	≈65%	Motor deficit, age, RM, recurrent tumor	Surgical ± aRT	-
Current study	27	75%	54%	RM	Surgical ± aRT	66.7%

Abbreviations: N = number of patients, 5 y DSS = 5-year disease-specific survival, 5 y DFS = 5-year disease-free survival, aRT = adjuvant radiotherapy, CHT = chemotherapy, RM = resection margins, WRCs = wound-related complications.

## Data Availability

All data supporting the results are available upon request, without prejudice to privacy or ethical restrictions.

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
