# Peer review of "Clinical Outcome after Surgical Treatment of Sacral Chordomas: A Single-Center Retrospective Cohort of 27 Patients"

_cancers, 2024, doi:10.3390/cancers16050973_

Round 1
Reviewer 1 Report
Comments and Suggestions for Authors
Thank you for allowing me to review this series of patients with a challenging problem. You present your series, and like other large centers the numbers of chordomas is small, but this data adds to the existing literature for us to say that we need to do better for these patients by finding ways to improve long term outcomes.
Specific Comments
- Table 2 is a little hard to understand. It would be better with only use whole numbers and %
- How many patients had total sacrectomies and how many were hemi?
- What about high (S2 and above) and low?
- There are classification schemes for these resections. Should you consider them? https://pubmed.ncbi.nlm.nih.gov/32941308/
- How many patients who had the internals ligated and had a VRAM had a wound complication? Were all the cases of internal ligation and wound complications seen in patients with gluteal flaps?
Comments on the Quality of English LanguageSome of the citing in the discussion is not correct
Author Response
Clinical Outcome After Surgical Treatment of Sacral Chordomas: A Single-Center Retrospective Cohort of 27 Patients
Stavros Goumenos, Georgios Kakouratos, Ioannis Trikoupis, Panagiotis Gavriil, Pavlos Gerasimidis, Konstantinos Soultanis, Pavlos Patapis, Vasileios Kontogeorgakos and Panayiotis Papagelopoulos
REVIEWER 1:
We kindly thank this reviewer for their constructive comments, through which we hope to enhance our manuscript, so that it meets the standards of the upcoming issue of your journal.
Specific Comments:
- Table 2 is a little hard to understand. It would be better with only use whole numbers and %.
Answer (1): We have added percentages for each respective subtotal sample in every row and column of both of our tables. Furthermore, there is a brief explanation of the data presentation format at the last row of each table.
- How many patients had total sacrectomies and how many were hemi?
Answer (2): All of our patients had either total or subtotal sacrectomies for their chordoma. None of them had a hemi-resection, because all of the tumors had crossed the midline in all cases (Lines 140-147).
- What about high (S2 and above) and low? and
- There are classification schemes for these resections. Should you consider them? https://pubmed.ncbi.nlm.nih.gov/32941308/
Answer (3,4): We adjusted the type and level of the sacrectomies, based on the proposed reference. For type 1C there is a more detailed description based on the level of the resection (Lines 140-147).
- How many patients who had the internals ligated and had a VRAM had a wound complication? Were all the cases of internal ligation and wound complications seen in patients with gluteal flaps?
Answer (5): All VRAM flaps and 3/5 gluteal flaps were mobilized in patients with internal iliac ligation (Line 138). All 4 partial flap necrosis occurred in patients with iliac vessel ligation (2 VRAM and 2 gluteal flaps, Lines 242-243). For all major wound-related complications regarding to type of flap mobilized, please refer to Table 2.
- Some of the citing in the discussion is not correct.
Answer 6: We reviewed the discussion part looking for mistakes regarding the citations, but did not find any except the number of patients (115) in line 323. If there is another mistake we failed to address, please let us know.
Reviewer 2 Report
Comments and Suggestions for Authors
Dear authors,
congratulations for this well performed research article on an important and interest topic.
I have several requests of improvements:
- 'Background': "Chordomas are a rare malignancy of notochordal origin accounting for 1-4% of primary bone tumors and for over 50% of all primary sacral tumors (1,2)." Do you refer to MALIGNANT bone and sacral tumors in the reported percentages? Please, check clarify and correct if necessary.
- 'Background': Please, enrich this section. I suggest you to add a paragraph in this section in regard to what is known about this topic. Discuss about other previous research focused on clinical outcome of patients treated with surgery for chordoma. Why is your research novel and/or important? What would you add to the existing knowledge of this topic? This to create a frame to your paper.
- 'Results' section "The overall survival rate during the entire follow-up in our cohort was 63% (17/27 172 patients)." Please, specify the mean follow-up time and range of the mentioned entire follow-up.
- 'Discussion' section line 2: "Tam" , please correct with "Team". Please, re-read your paper with attention and check for typos before re-submission.
- 'Figure Legends' Please, adjust references with the correct radiological terms. All the MRI sequences showed must be mentioned correctly in each panel.
- 'Tables'. Tables and text seems to be partially superimposed. This is probably a problem that the editorial team/office should correct. Anyway, check it out.
- 'Tables'. I suggest you adding a table summarizing the other papers (most improtant-largest ones) previously published on treatment of chordoma including in rows n. of patients included, treatments (surgery, surgery + RT, RT), mean survival, other notes..
Comments on the Quality of English LanguageQuality of english is overall very good.
Author Response
Clinical Outcome After Surgical Treatment of Sacral Chordomas: A Single-Center Retrospective Cohort of 27 Patients
Stavros Goumenos, Georgios Kakouratos, Ioannis Trikoupis, Panagiotis Gavriil, Pavlos Gerasimidis, Konstantinos Soultanis, Pavlos Patapis, Vasileios Kontogeorgakos and Panayiotis Papagelopoulos
REVIEWER 2:
We kindly thank this reviewer for their constructive comments, through which we hope to enhance our manuscript, so that it meets the standards of the upcoming issue of your journal.
Specific Comments:
- 'Background': "Chordomas are a rare malignancy of notochordal origin accounting for 1-4% of primary bone tumors and for over 50% of all primary sacral tumors (1,2)." Do you refer to MALIGNANT bone and sacral tumors in the reported percentages? Please, check clarify and correct if necessary.
Answer: The reference was regarding malignant tumors. We corrected this part in lines 56-57: “Chordomas are a rare malignancy of notochordal origin accounting for 1-4% of malignant bone tumors and for over 50% of all primary tumors of the sacrum.
- 'Background': Please, enrich this section. I suggest you to add a paragraph in this section in regard to what is known about this topic. Discuss about other previous research focused on clinical outcome of patients treated with surgery for chordoma. Why is your research novel and/or important? What would you add to the existing knowledge of this topic? This to create a frame to your paper.
Answer: We enriched the Introduction part with a more suitable paragraph (Lines: 64-69).
- 'Results' section "The overall survival rate during the entire follow-up in our cohort was 63% (17/27 172 patients)." Please, specify the mean follow-up time and range of the mentioned entire follow-up.
Answer: The mean follow-up was added in this part (Line: 191).
- 'Discussion' section line 2: "Tam" ,please correct with "Team". Please, re-read your paper with attention and check for typos before re-submission.
Answer: The term was replaced in line 284. The manuscript was checked again for other mistakes. If there is another mistake we failed to address, please let us know.
- 'Figure Legends' Please, adjust references with the correct radiological terms. All the MRI sequences showed must be mentioned correctly in each panel.
Answer: The figure legends were adjusted with the correct MRI views and sequences (Lines: 117-120, 128-135 and 149-156 respectively).
- Tables'. Tables and text seems to be partially superimposed. This is probably a problem that the editorial team/office should correct. Anyway, check it out.
Answer: We modified the text format in accordance with the size and position of the tables and we will contact the editorial office concerning this matter.
- 'Tables'. I suggest you adding a table summarizing the other papers (most improtant-largest ones) previously published on treatment of chordoma including in rows n. of patients included, treatments (surgery, surgery + RT, RT), mean survival, other notes..
Answer: We added a table (Table 3) in the Discussion part (Line 407) describing the results of some selected large cohorts about chordomas of the sacrum.
Round 2
Reviewer 1 Report
Comments and Suggestions for Authors
Thank you for making the corrections we requested. I have no further comments/concerns
Comments on the Quality of English LanguageMinor grammar issues, nothing major
Author Response
We kindly thank the reviewer for his constructive comments, through which we enhanced our manuscript, so that it meets the standards of the upcoming issue of your journal. All comments have been addressed.
Reviewer 2 Report
Comments and Suggestions for Authors
Dear authors,
I am partially satisfied about the revisions performed,
anyway there are still some concerns to fix in regards to figure legends (MRI parts):
- Figure 2 part B this is not a T1w sequence. part C this is not a T1w coronal. part I and J this is not a T1weighted
- Figure 3 A and B this is not a T1w sequence.
It is necessary to involve a radiologist who check and correct the part in regards to MRI legends.
Comments on the Quality of English LanguageMinor language inaccuracies and typos should be corrected.
Author Response
We kindly thank the reviewer for their constructive comments, through which we enhanced our manuscript, so that it meets the standards of the upcoming issue of your journal. Regarding the specific comment about the MRI sequences illustrated, we have consulted a specialized radiologist and corrected all the figure legends as recommended. Thank you again for improving our manuscript.